# Role of Obesogens in the Pathogenesis of Obesity

**DOI:** 10.3390/medicina55090515

**Published:** 2019-08-21

**Authors:** Urszula Shahnazaryan, Marta Wójcik, Tomasz Bednarczuk, Alina Kuryłowicz

**Affiliations:** Department of Internal Diseases and Endocrinology, Medical University of Warsaw, 1a Banacha St, 02-097 Warsaw, Poland

**Keywords:** obesogens, obesity, adipose tissue, endocrine-disrupting chemicals, metabolism disrupting chemicals

## Abstract

Obesity is considered to be a 20th century pandemic, and its prevalence correlates with the increasing global pollution and the presence of chemical compounds in the environment. Excessive adiposity results from an imbalance between energy intake and expenditure, but it is not merely an effect of overeating and lack of physical activity. Recently, several compounds that alter the mechanisms responsible for energy homeostasis have been identified and called “obesogens”. This work presents the role of obesogens in the pathogenesis of obesity. We reviewed data from in vitro animal and human studies concerning the role of obesogens in the disturbance of energy homeostasis. We identified (i) the main groups and classes of obesogens, (ii) the molecular mechanisms of their action, (iii) their deleterious effect on adipose tissue function and control of appetite, and (iv) possible directions in limiting their influence on human metabolism. Obesogens have a multifactorial detrimental influence on energy homeostasis. Focusing on limiting exposure to obesogens and improving early life nutrition seems to be the most reasonable direction of action to prevent obesity in future generations.

## 1. Introduction

The increasing number of overweight and obese individuals in the world (and the related health and socioeconomic consequences) has forced researchers to look for the potential reasons for this pandemic. Obesity is a result of an imbalance between energy intake and expenditure [1,2,3,4], but it is not merely an effect of overeating and lack of physical activity. Excessive body mass gain is the result of a prolonged disturbance in the hormonally controlled homeostasis of energy balance, in which the metabolisms of many tissues and organs (gastrointestinal tract, pancreas, muscle, adipose tissue, liver, and brain) are involved. Processes such as appetite control, accumulation and mobilization of lipids from adipose storage depots, and the basal metabolic rate depend on a variety of hormonal factors [5], which in turn interact with human behavior, genetic predisposition, and the environment.

The current increase in worldwide obesity correlates with an increase in exposure to environmental pollution. Based on this finding, in 2004, Gluckman and Hanson proposed the “Developmental Origins of Health and Disease” (DOHaD) model, which established that life-long patterns of health and disease depend on interactions between genetic factors and exposure to environmental factors in early life [6]. Later, the same investigators suggested that events occurring during the prenatal period are of critical importance for metabolism and result in permanent changes that manifest in adolescence and adulthood [7]. In parallel, other researchers have focused on factors occurring during the postnatal period that could lead to a predisposition to obesity, finding a link between global pollution, the chemicals present in the environment, and irreversible metabolic disturbances [8]. Environmental factors that may contribute to the development of obesity are summarized in Table 1.

Baille-Hamilton was the first to suggest that the current obesity epidemic is associated with the increase in environmental pollution from chemicals. In 2002, she published a paper entitled “Chemical toxins: a hypothesis to explain the global obesity epidemic”, in which she reported an association between weight gain of laboratory animals and their exposure to various chemical compounds (pesticides, solvents, plastics, flame retardants, and heavy metals) [8]. Other researchers have focused primarily on endocrine-disrupting chemicals (EDCs) as the most suspected substances. EDCs are a large group of different chemical compounds that “interfere with the production, release, transport, metabolism, binding, action, or elimination of natural hormones in the body responsible for the maintenance of homeostasis and the regulation of developmental processes” (US EPA definition 1996) [9,10]. Presently, more than 1300 substances have been recognized as potentially able to interfere with endocrine homeostasis [11]. Some of them are quickly degraded, but many undergo bioaccumulation. Table 2 shows the main groups of EDCs and examples of each one.

EDCs that target metabolic “set-points” and, in this way, create a predisposition to weight gain are known as obesogens. This term was first introduced by Blumberg in reference to substances that alter mechanisms responsible for lipid homeostasis [12,13]. Subsequently, obesogens, by altering the pathways of food intake and energy metabolism, were found to promote obesity directly, by increasing the number of fat cells and/or the storage of fat in existing adipocytes, and indirectly, by shifting the energy balance to favor calorie storage (by altering the basal metabolic rate to promote food storage) and affecting the hormonal control of appetite and satiety [14,15].

In this review, we present the molecular mechanisms of obesogen action and the evidence regarding their contribution to adipose tissue dysfunction as well as to the collapse of mechanisms responsible for appetite control, which ultimately promote obesity.

## 2. Sources and Classes of Obesogens

To date, about 20 different chemical compounds have been identified as obesogens. Some of them are present in nature (e.g., phytoestrogens), while most are synthetic chemical compounds that have been intentionally or unintentionally released into the environment [16]. Exposure to these substances may occur orally, via inhalation, or by dermal absorption.

One of the best documented groups of chemicals with obesogenic properties are pesticides, in particular, organotins such as tributyltin oxide (TBT) and triphenyltin (TPT) [17]. Polycyclic aromatic hydrocarbons (PAHs), including benzo[*a*]pyrene, are byproducts of fuel burning [18]. Bisphenol A (BPA) is a compound present in polycarbonate plastics, which are widely used in products such as beverage containers and food cans [19]. Polybrominated biphenyls and polybrominated diphenyl ethers (PBDEs) are used as flame retardants [20]. Another group of obesogens are phthalates, which are diesters of phthalic acid used to impart flexibility to plastic products and as carriers for fragrances in cosmetics. They can easily leach from these products and can be found in indoor air and house dust [21].

Obesogenic properties have been associated with alkylphenols (derivatives of alkylphenol ethoxylates), which are nonionic surfactants that are used as cleaning agents [22]. There is also a body of evidence for the obesogenic effects of the pesticide dichlorodiphenyltrichloroethane (DDT) and its metabolite dichlorodiphenyldichloroethylene (DDE) [23]. Other representative obesogens are parabens present in antimicrobial agents used for the preservation of food, paper, and pharmaceutical products [24]. Groups of obesogens are listed in Table 3. 

## 3. Mechanisms of Obesogen Action

Obesogens exert their effects via reprogramming different signaling pathways that have common endpoints in tissues crucial for whole-body metabolism, resulting in increased adiposity and/or altered function of adipose tissue.

### 3.1. Obesogens Acting as Receptors’ Ligands and Transcription Factors

Blumberg, during his early experiments, expected TBT to activate nuclear sex steroid receptors, but instead, he observed activation of the peroxisome proliferator-activated receptor gamma (PPARγ) [17]. PPARγ is a crucial receptor for adipogenesis and functioning of the mature adipocyte. Its activation stimulates mesenchymal stem cells (MSCs) to differentiate towards adipocytes (not to bone cells) and is responsible for the initiation of lipogenesis [36]. Apart from TBT, many other obesogens target PPARγ by increasing its expression or binding to it directly to activate downstream cascades that lead to enhanced adipogenesis. These include, among others, DDT and its metabolite DDE, nonylphenol (NP), octylphenol (OP), BPA, di-(2-ethylhexyl)phthalate (DEHP), dibutyl phthalate (DBP), benzyl butyl phthalate (BBP), and mono-benzyl phthalate (MBzP) [26]. Aside from affecting PPARγ, obesogens may directly increase the expression of genes involved in lipid storage in adipocytes, for example, lipoprotein lipase and fatty-acid-binding protein 4/adipocyte protein 2 (aP2). These phenomena occur in response to DDT, DDE, 4NP, BPA, DEHP, mono(2-ethylhexyl) phthalate (MEHP), and BBP (reviewed in [26]).

Moreover, BPA was found to modify adipose tissue metabolism by influencing the function of glucocorticoid receptors (GRs). BPA-binding GRs directly increase lipid accumulation and adipogenesis and do so indirectly by increasing mRNA expression and the enzymatic activity of 11beta-hydroxysteroid dehydrogenase 1 (an enzyme converting cortisone to cortisol) [27].

Classically, activation of estrogen receptors (ERs) in MSCs inhibits adipogenesis. However, obesogens (e.g., DDT, NP, and BPA) can activate ER-mediated signaling in MSCs in such a way that it enhances adipogenesis [28]. Interestingly, some obesogens (e.g., TBT and alkylphenols) were found to modulate the activity of aromatase (CYP19A1), which is a key enzyme in estrogen synthesis responsible for the conversion of androgens to estrone (E1) or estradiol (E2) [25].

Phthalates can also interfere with the thyroid hormone system, which is critical to maintaining the basal metabolic rate. The antithyroid properties of phthalates have been shown both in vivo and in vitro [31], and epidemiological studies have reported a negative association between urinary phthalate metabolite concentrations and thyroid hormones, as well as testosterone levels [32].

### 3.2. Influence of Obesogens on Epigenetic Modifications

Out of the many potential mechanisms regulating gene expression in adipose tissue, epigenetic modifications have been of particular interest in recent years. Epigenetics is defined as changes in gene function, such as DNA methylation, histone modifications (acetylation, methylation, phosphorylation, or ubiquitination), and microRNA (miRNA) interference, that occur without a change in the DNA sequence. There is cumulative evidence that exposure to obesogens during development via, for example, alterations in DNA methylation, histone acetylation, and miRNA expression, can lead to persistent changes in gene activities in tissues crucial for the regulation of metabolism.

Depending on the applied concentration, the phthalate BBP was shown to cause histone modifications that induce MSCs to undergo adipogenic differentiation. These include decreased methylation of PPARγ, enhancement of H3K9 acetylation, an increase of histone acetyltransferase expression along with a reduction of histone deacetylase expression, and decreased dimethylation of H3K9 [33]. Pregnant mice that came into contact with PAH resulted in lower DNA methylation of PPARγ in their offspring [18]. In turn, exposure of different cells to BPA resulted in decreased trimethylation of histone H3K9 and increased expression of miR-146a; however, this has not yet been shown in MSCs [29].

## 4. Influence of Obesogens on Adipose Tissue Development, Metabolism, and Appetite Control

### 4.1. Adipogenesis

Obesogens can increase the amount of adipose tissue by increasing the number and size of adipocytes. The ability to induce differentiation of MSCs into preadipocytes and subsequently to adipocytes (in the mechanisms described above) was shown in vitro, for example, for TBT, DDT, NP, OP, BPA, PCB, DEHP, MEHP, DBP, BBP, MBzP, and parabens [26,37]. However, obesogens can also influence adipogenesis in a paracrine manner by affecting the intercellular milieu of preadipocytes or MSCs. For instance, PCBs, by inhibiting the release and action of leptin, can induce adipogenesis and promote fatty acid storage to form triglycerides [38]. Nevertheless, the results of in vitro studies should be treated with caution since, in one experiment, the influence of only one obesogen was assessed and there is a lack of data on the effects of MSCs being exposed to multiple obesogens simultaneously. It might be that interactions between different obesogens may lead to an additive effect on adipocyte proliferation.

The influence of obesogens on preadipocyte differentiation was also demonstrated in several in vivo studies, both in vertebrata and invertebrata, as well as in some human studies. Exposure of pregnant mice to TBT predisposed their offspring to have higher adipose tissue content than of those which were not exposed [17]. Similar observations were made regarding animals in puberty and early adulthood [39]. In turn, exposure in utero and early childhood to PBDEs resulted in increased weight gain both in experimental animals and in children and is associated with thyroid dysfunction and altered testosterone metabolism [31,32,40]. DEHP treatment on father or mother *Drosophila melanogaster* resulted in increased or decreased body weight of the offspring, respectively [41]. Prenatal and neonatal exposure to diethylhexyl phtalate in mice led to an increased number of adipocytes and subsequently to higher body weight in offspring and adult animals [34,35]. In turn, in epidemiological studies, increased urinary concentrations of phthalate metabolites were correlated with higher waist circumference and body mass index (BMI) [42]. Similarly, prenatal exposure to DDT increased the adiposity of subsequent generations of rodents, while in epidemiological studies, prenatal exposure to DDT and DDE increased the risk of human obesity [23].

In epidemiological studies, maternal exposure to BPA resulted in low birth weight, which is a known risk factor for obesity in adult life, while urinary BPA levels correlated positively with BMI and waist circumference in children and adults [43]. However, the results of other animal and human studies regarding a possible link between exposure to BPA (and its analogues) and the development of obesity are inconclusive and further research is required to clarify these mechanisms.

### 4.2. Adipose Tissue Metabolism

Cell culture studies have shown that many obesogens not only induce preferential differentiation of MSCs into preadipocytes but also alter the metabolism of mature adipocytes so that they become dysfunctional (reviewed in [44,45]).

3T3-L1 cells differentiated with TBT had a greater ability to accumulate triglycerides and reduced expression of glucose transporter 4 (GLUT4) [46]. The TBT-treated cells also had fewer mitochondria, a lower respiration rate, and a lower potential for browning [47]. This is consistent with the fact that mice exposed to TBT in the perinatal period have a greater potential to accumulate adipose tissue on a high-fat diet and impaired ability to mobilize these depots under fasting conditions [48]. The lipogenic effect of TBT is not limited to mammals, since it was found to modulate the transcription of key genes regulating lipid metabolism and enzymes involved in lipogenesis in the liver of exposed zebrafish (*Danio rerio*) [49]. TBT also impairs the transfer of triacylglycerols to eggs of *Daphnia magna* and hence promotes their accumulation in adult individuals, which translates to lower fitness for the offspring and adults [50]. Similarly, BPA analogues induce lipid accumulation in zebrafish larvae and late-onset weight gain in juvenile zebrafish [51].

Prenatal and neonatal exposure to diethylhexyl phtalate in mice, via modulation of the transcription of genes involved in lipid metabolism, increased the number and size of adipocytes and subsequently led to higher body weight in offspring and adult animals, which was confirmed in cell lines [39,40]. When added to adipocyte cell cultures, alkylphenols were also found to alter transcription patterns of several genes involved in lipid metabolism, and their adipogenic potency increased with the increasing length of the linear alkyl chain [22,37] Similarly, in vivo, exposure of rats to NP resulted in their increased adiposity [52].

Moreover, in adipose tissue, obesogens can promote an inflammatory state and increase oxidative stress. Oxidative stress results from an imbalance in the production of reactive oxygen species (ROS) and their detoxification by antioxidants and is associated with damage and early senescence of cells. Increased ROS levels are associated with preferential differentiation of MSCs towards adipocytes. Some obesogens (e.g., DEHP and BPA) have been shown to promote oxidative stress and increase ROS levels in mature adipocytes, which results in enhanced lipid oxidation in different vertebrates [26,53]. Also, alkylphenols have been shown to significantly induce the generation of ROS and inhibit placental aromatase activity [22], which results in increased lipid accumulation and estrogenic effects.

Some obesogens (e.g., BPA and PCBs) increase the levels of proinflammatory cytokines, such as TNF-α and interleukin 6, in cell cultures of adipocytes, inducing a proinflammatory phenotype in adipose tissue [26]. An increase in proinflammatory cytokine levels in adipose tissue following exposure to parathion and PCB-77 was also observed in animal studies [54]. The long-term exposure of MSCs to obesogenic compounds (e.g., BPA and PCBs) results in a proinflammatory state that can inhibit osteogenesis and, in this way, promote differentiation towards adipocytes [55]. Therefore, obesogens may contribute to the development of metaflammation, a low-grade chronic inflammatory state observed in obese individuals which is believed to constitute a link between obesity and related complications, including cardiovascular disease, type 2 diabetes mellitus, and dyslipidemia.

### 4.3. Regulation of Appetite, Satiety, and Food Preference

Obesogens may also influence the function of the hypothalamus, which is the region of the brain responsible for controlling eating behavior. It was shown in rats that exposure to BPA in early life alters both presynaptic and postsynaptic signaling pathways, promoting addictive and compulsive behavior and resulting in increased food consumption and subsequent obesity [56].

Moreover, obesogenic compounds can also influence appetite by modulation of adipokine secretion. Exposure of 3T3L1 adipocytes to BPA increases the production of leptin mRNA levels [19]. Similarly, exposure of mice neonates to methylparaben increases the serum leptin level, while exposure to diethylhexyl phtalate increases leptin and decreases adiponectin levels, subsequently leading to higher body weight in offspring and adult animals [34,35]. In humans, BPA serum levels correlate with serum levels of leptin, ghrelin levels, and body mass [30]. All these data suggest that, by interfering with adipokine secretion, obesogens may affect the hormonal control of hunger and satiety.

## 5. Susceptibility to Obesogens

Organisms differ in their susceptibility to obesogens. The most susceptible to their negative influence are fetuses and newborns. This is due to the lack of physiological protective mechanisms in immature organisms which are fully functioning in adults (e.g., DNA repair ability, liver metabolism and detoxifying enzyme action, a competent immune system, and the blood/brain barrier). The effects of obesogens are exerted by disrupting the epigenetic signaling (e.g., as mentioned above, DNA methylation, histone marks, chromatin remodeling, and miRNAs) that regulates gene expression during development [57]. Changes in these mechanisms lead to permanent abnormal gene expression patterns, are responsible for the deleterious effects of obesogens, and consequently, contribute to the development of many metabolic diseases across the life course [58]. The exclusive obesogenic influence of different compounds on fetuses and newborns has been demonstrated in several animal models. For instance, dietary phytoestrogens (genistein and daidzein), via modulation of ER signaling, can induce apoptosis in preadipocytes in vitro and reverse abdominal accumulation of adipose tissue in ovariectomized rodents and postmenopausal women [59]. However, exposure to phytoestrogens in the fetal or neonatal period has the opposite effect: in rodents, it leads to obesity at puberty [16].

It is also suggested that exposure to obesogens in early life might have a more significant influence than genetic predisposition on metabolic disease development. Moreover, the pathomechanism acts as a vicious cycle arising from the fact that most obesogens have lipophilic properties—the more adipose tissue in the body, the more significant the reservoir for obesogens in the organism. The final effect of exposure to obesogens varies across individuals because it depends on the dose, route of exposure, and co-occurrence of many other environmental stressors, such as low socioeconomic status, stress, sleep disturbances, anxiety, depression, drugs, hypercaloric diet, activity, infections, microbiome, and so forth. As a rule, exposure is not limited to a single compound but to mixtures of chemicals. Therefore, establishing the environmental effects of each case requires knowledge of the target sensitivities, the dose–response relationship, and the pharmacokinetics of each compound (half-lives, metabolism, tissue accumulation, and persistence). Evaluation of the role of obesogens in each individual includes sensitive personal measurements to identify the phenotype. This is difficult due to human genomic variability, which creates population diversity. Because some effects may be transgenerational, this approach may require data from a minimum of two generations of the paternal line and three generations of the maternal line [60].

## 6. Final Remarks and Conclusions

The concept of obesogens and the theory of metabolic programming by chemicals provide a plausible answer to the question which the “calories in–calories out” model cannot yet explain—Why are there such vast differences in individuals’ intensity of appetite, sensitivity to physical activity, and facility to maintain body weight?

Given the accumulated data about the role obesogens play in adipose tissue development, metabolism, and secretory activity, which subsequently influence whole body function, a multidisciplinary group of experts gathered in Parma, Italy elaborated the Parma Consensus Statement [60]. They proposed to broaden the definition of obesogens to include metabolism-disrupting chemicals (MDCs) in order to encompass chemicals that play a role in altered susceptibility to obesity, diabetes, and related metabolic disorders such as metabolic syndrome. The Parma Consensus Statement also suggested directions for further research, which include the characterization of the pathways through which MDCs act; identification of windows and mechanisms of susceptibility; and establishment of multiple endpoints, differences in sex and early biomarkers that could be used to predict disease outcomes later in life.

## Figures and Tables

**Table 1 medicina-55-00515-t001:** Environmental Factors Contributing to Obesity Development (Based on [8]).

Environmental Factors Contributing to Obesity Development
**Prenatal exposure**
Environmental toxins (obesogens)
Viruses
Maternal diet with a high glycemic index
**Postnatal exposure**
Environmental toxins (obesogens)
The increasing glycemic index of food
Lack of physical activity
Watching television and playing computer games, which results in: - displacement of physical activity - depression of metabolic rate - adverse effects on diet quality - effects of television on sleep
Shortened sleep duration
Medications, e.g., - neuroleptics - antiepileptics - glucocorticoids
Disturbances of gut microbiota

**Table 2 medicina-55-00515-t002:** The main groups of endocrine-disrupting chemicals (EDCs) and examples.

Group of Substances	Example(s)
Synthetic hormones	Ethynylestradiol
Plastics	bisphenol A (BPA) phthalates
Pesticides Fungicides	organotins methoxychlor chlorpyrifos dichlorodiphenyltrichloroethane (DDT) vinclozolin
Solvents	polychlorinated biphenyls (PCBs) polybrominated biphenyls (PBBs) dioxins
Pharmaceutical agents	thiazolidinediones atypical antipsychotics antihistamines antidepressants
Personal care products	triclosan
Phytoestrogens	genistein coumestrol

**Table 3 medicina-55-00515-t003:** Groups of obesogens, including their role in the environment, mechanism of action, and endpoints observed in cell lines, rodents, and humans.

Group	Name of the Chemical Compound	Role/Source in the Environment	Mechanism of Action	Endpoints
organotins	tributyltin oxide (TBT) triphenyltin (TPT)	used as a biocide (fungicide and molluscicide), especially as a wood preservative [17]	- peroxisome proliferator-activated receptor gamma (PPARγ) and retinoid X receptor alpha (RXRα) activators [17] - inhibitors of CYP19 [25]	↑adipogenesis ↑lipids storage ↓synthesis of estrogens
polycyclic aromatic hydrocarbons (PAHs)	Benzo[*a*]pyrene	byproducts of fuel burning [18]	- PPARγ activation via epigenetic modifications [18]	↑adipogenesis ↑lipids storage
bisphenol A (BPA) and its analogues	tetrabromobisphenol A (TBBPA) tetrachlorobisphenol A (TCBPA)	components of polycarbonate plastics [19]	- PPARγ activators [26] - glucocorticoid and estrogen receptor modulators [27,28] - direct activators of lipogenic genes [26] - epigenetic modifications [29] -↑generation of reactive oxygen species (ROS) [26] - ↑ inflammation [26] - ↑leptin level [19,30]	↑lipids storage ↑adipogenesis ↑appetite
polybrominated diphenyl ethers (PBDEs)	lower brominated PBDEs (with 1–4 bromine atoms)	flame retardants [20]	- interference with thyroid function [31,32] and testosterone metabolism [32]	↑lipids storage
phthalates	di-(2-ethyl hexyl)phthalate (DEHP), dibutyl phthalate (DBP), mono-benzylphthalate (MBzP), benzyl butyl phthalate (BBP), mono(2-ethylhexyl) phthalate (MEHP)	plasticizing agents present in cosmetics, paints, and medicines [21]	- PPARγ activators [26] - direct activators of lipogenic genes [28] - epigenetic modifications [33] -↑generation of ROS [26] -↑leptin level -↓adiponectin level [34,35]	↑adipogenesis ↑lipids storage ↑appetite
alkylphenols	nonylphenol (NP) octylphenol (OP)	surfactants in industrial and consumer products [22]	- PPARγ activators [26] - direct activators of lipogenic genes [26] -estrogen receptor modulators [28] - inhibitors of CYP19 [29] -↑generation of ROS [22]	↑lipids storage ↑adipogenesis ↓synthesis of estrogens
DDTDDE	dichlorodiphenyltrichloroethane dichlorodiphenyldichloroethylene	pesticides [23]	- PPARγ activators [26] - direct activator of lipogenic genes [26] - estrogen receptor modulators [28]	↑lipids storage ↑adipogenesis
parabens	alkyl esters of p-hydroxybenzoic acid	antimicrobial agents for the preservation of food, paper products, and pharmaceutical products [24]	- PPARγ activators [26] - glucocorticoid and estrogen receptor modulators [28] -↑leptin level [34,35]	↑lipids storage ↑adipogenesis
phytoestrogens	isoflavones	components of food (genistein, daidzein in soybeans, legumes, lentils, chickpeas) [16]	- estrogen receptor modulators [16]	↑lipids storage ↑adipogenesis

↑ increase, ↓ decrease.

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
