# Peer review of "Role of Obesogens in the Pathogenesis of Obesity"

_medicina, 2019, doi:10.3390/medicina55090515_

Round 1
Reviewer 1 Report
The article is well-written and adds to the existing literature on this important topic. I don't have any comments.
Author Response
We would like to express our gratitude to the Reviewer for the generally positive reception of the manuscript.
Reviewer 2 Report
This manuscript provides a comprehensive consideration of the role of obesogens in the pathogenesis of obesity. The order of the topics covered was appropriate. In addition, the detail considered in each topic area was appropriate.
I have no concerns, other than minor revisions to increase fluency of the narrative, with this manuscript. If I were required to find one critique it would be that only mammalian systems have been considered. To provide some additional information concerning the role of obesogens in vertebrate models, such as zebrafish, and invertebrate models, such as fruit flies would have been appreciated.
Author Response
We would like to express our gratitude to the Reviewer for the generally positive reception of the manuscript.
I have no concerns, other than minor revisions to increase fluency of the narrative, with this manuscript.
Following the Reviewer suggestion, we took the advantage from the English editing service provided by MDPI (see the attached certificate).
If I were required to find one critique, it would be that only mammalian systems have been considered. To provide some additional information concerning the role of obesogens in vertebrate models, such as zebrafish, and invertebrate models, such as fruit flies would have been appreciated.
Indeed, writing the manuscript, we had focused on the contribution of obesogens to development of obesity only in mammals, neglecting the data from other vertebrate and invertebrate models. Therefore, in the revised version of the manuscript, we introduced some examples of studies regarding the influence of obesogens on lipid metabolism and weight gain in, e.g., zebrafish, Drosophila melanogaster and Daphnia magna.
“The influence of obesogens on preadipocyte differentiation was also demonstrated in several in vivo studies, both in vertebrata and invertebrata, as well as in some human studies.” Page 6, lines 162-163.
“DEHP treatment on father or mother Drosophila melanogaster resulted in increased or decreased bodyweight of the offspring, respectively [38].” Page 6, lines 168-169.
“The lipogenic effect of TBT is not limited to mammals, since it was found to modulate the transcription of key genes regulating lipid metabolism and enzymes involved in lipogenesis in the liver of exposed zebrafish (Danio rerio) [48]. TBT also impairs the transfer of triacylglycerols to eggs of Daphnia magna and hence promotes their accumulation in adult individuals, which translates to lower fitness for the offspring and adults [49]. Similarly, BPA analogues induce lipid accumulation in zebrafish larvae and late-onset weight gain in juvenile zebrafish [50].” Page 6, lines 191-196.
“Some obesogens (e.g., DEHP and BPA) have been shown to promote oxidative stress and increase ROS levels in mature adipocytes, which results in enhanced lipid oxidation in different vertebrates [26,52].” Page 6, lines 208-210.
We also added proper citations to the References section.
Chen, M.Y., Liu, H.P., Cheng, J., Chiang, S.Y., Liao, W.P., Lin, W.Y. Transgenerational impact of DEHP on body weight of Drosophila. Chemosphere 2019, 221, 493-499. Available online: https://www.sciencedirect.com/science/article/pii/S0045653518325359/ (accessed on 09.08.2019).Lyssimachou, A., Santos, J.G., André, A., Soares, J., Lima, D., Guimarães, L., Almeida, C.M., Teixeira, C., Castro, L.F., Santos, M.M. The Mammalian "Obesogen" Tributyltin Targets Hepatic Triglyceride Accumulation and the Transcriptional Regulation of Lipid Metabolism in the Liver and Brain of Zebrafish. PLoS One 2015, 10, e0143911. Available online: https://journals.plos.org/plosone/article?id=10.1371/journal.pone.0143911/ (accessed on 09.08.2019).
Jordão, R., Casas, J., Fabrias, G., Campos, B., Piña, B., Lemos, M.F., Soares, A.M., Tauler, R., Barata, C. Obesogens beyond Vertebrates: Lipid Perturbation by Tributyltin in the Crustacean Daphnia magna. Environ Health Perspect 2015, 123, 813-819 Available online: https://ehp.niehs.nih.gov/doi/pdf/10.1289/ehp.1409163/(accessed on 09.08.2019).
Riu, A., McCollum, C.W., Pinto, C.L., Grimaldi, M., Hillenweck, A., Perdu, E., Zalko, D., Bernard, L., Laudet, V., Balaguer, P., Bondesson, M., Gustafsson, J.A. Halogenated bisphenol-A analogs act as obesogens in zebrafish larvae (Danio rerio). Toxicol Sci 2014 139, 48-58. Available online: https://academic.oup.com/toxsci/article/139/1/48/2338284/ (accessed on 09.08.2019).
Canesi, L., Fabbri, E. Environmental Effects of BPA: Focus on Aquatic Species. Dose Response 2015, 13, 1559325815598304. Available online: https://www.ncbi.nlm.nih.gov/pmc/articles/PMC4674185/ (accessed on 09.08.2019).
